

# An inter-comparison of total column-averaged nitrous oxide between ground-based FTIR TCCON and NDACC measurements at seven sites and comparisons with the GEOS-Chem model

Minqiang Zhou[1], Bavo Langerock[1], Kelley C. Wells[2], Dylan B. Millet[2], Corinne Vigouroux[1], Mahesh Kumar Sha[1], Christian Hermans[1], Jean-Marc Metzger[3], Rigel Kivi[4], Pauli Heikkinen[4], Dan Smale[5], David F. Pollard[5], Nicholas Jones[6], Nicholas M. Deutscher[6], Thomas Blumenstock[7], Matthias Schneider[7], Mathias Palm[8], Justus Notholt[8], James W. Hannigan[9], and Martine De Mazière[1]

[1]Royal Belgian Institute for Space Aeronomy (BIRA-IASB), Brussels, Belgium
[2]Department of Soil, Water, and Climate, University of Minnesota, St. Paul, MN, USA
[3]UMS 3365 – OSU Réunion, Université de La Réunion, Saint-Denis, Réunion, France
[4]Finnish Meteorological Institute, Space and Earth Observation Centre, Sodankylä, Finland
[5]National Institute of Water and Atmospheric Research, Lauder, New Zealand
[6]Centre for Atmospheric Chemistry, University of Wollongong, Wollongong, Australia
[7]Institute of Meteorology and Climate Research, Karlsruhe Institute of Technology, Karlsruhe, Germany
[8]Institute of Environmental Physics, University of Bremen, Bremen, Germany
[9]Atmospheric Chemistry Observations and Modeling, National Center for Atmospheric Research, Boulder, CO, USA

**Correspondence:** Minqiang Zhou (minqiang.zhou@aeronomie.be)

**Abstract.** Nitrous oxide ($N_2O$) is an important greenhouse gas and it can also generate nitric oxide, which depletes ozone in the stratosphere. It is a common target species of ground-based FTIR near-infrared (TCCON) and mid-infrared (NDACC) measurements. Both TCCON and NDACC networks provide a long-term global distribution of atmospheric $N_2O$ mole fraction. In this study, the dry-air column averaged mole fraction of $N_2O$ ($X_{N_2O}$) from the TCCON and NDACC measurements

are compared against each other at seven sites around the world (Ny-Ålesund, Sodankylä, Bremen, Izaña, Reunion Island, Wollongong, Lauder) in the time period of 2007-2017. The mean differences in $X_{N_2O}$ between the TCCON and NDACC (NDACC-TCCON) at these sites are between -3.32 and 1.37 ppb (-1.1 – 0.5 %) with the standard deviations between 1.69 and 5.01 ppb (0.5 – 1.6 %), which are within the uncertainties of the two datasets. The NDACC $N_2O$ retrieval has good sensitivity throughout the troposphere and stratosphere, while the TCCON retrieval underestimates a deviation from the a priori in the

troposphere and overestimates it in the stratosphere. As a result, the TCCON $X_{N_2O}$ measurement is strongly affected by its a priori profile.

Trends and seasonal cycles of $X_{N_2O}$ are derived from the TCCON and NDACC measurements and the nearby surface flask sample measurements, and compared with the results from GEOS-Chem model a priori and a posteriori simulations. The a posteriori $N_2O$ fluxes in the model are optimized based on surface $N_2O$ measurements with a 4D-Var inversion method.

The $X_{N_2O}$ trends from the GEOS-Chem a posteriori simulation are very close to those from the NDACC and the surface flask sample measurements (0.9 – 1.0 ppb/year). The $X_{N_2O}$ trends from the TCCON measurements are slightly lower (0.8 – 0.9 ppb/year) due to the underestimation of the trend in TCCON a priori. The $X_{N_2O}$ trends from the GEOS-Chem a priori



simulation are about 1.25 ppb/year, and our study confirms that the N$_2$O fluxes from the a priori inventories are overestimated. The seasonal cycles of X$_{N_2O}$ from the FTIR measurements and the model simulations are close to each other in the Northern Hemisphere with a maximum in August-October and a minimum in February-April. However, in the Southern Hemisphere, the modeled X$_{N_2O}$ shows a minimum in February-April while the FTIR X$_{N_2O}$ retrievals shows a minimum in August-October.

By comparing the partial column averaged N$_2$O from the model and NDACC for three vertical ranges (surface–8, 8–17, 17–50 km), we find that the discrepancy in the X$_{N_2O}$ seasonal cycle between the model simulations and the FTIR measurements in the Southern Hemisphere is mainly due to their stratospheric differences.

# 1 Introduction

Nitrous oxide (N$_2$O) is the third most important anthropogenic greenhouse gas in the Earth's atmosphere after carbon dioxide

(CO$_2$) and methane (CH$_4$) (IPCC, 2013). In addition, N$_2$O is a precursor of ozone depleting nitric oxide radicals and it is an important anthropogenic cause of stratospheric ozone depletion (Ravishankara et al., 2009; Portmann et al., 2012). The globally averaged N$_2$O mole fraction in the atmosphere was 328.9 ppb (part per million volume) in 2016, representing a 22% increase since 1750. The annual growth rate of N$_2$O in the last decade is about 0.90 ppb/year derived from direct National Oceanic and Atmospheric Administration - Global Monitoring Division (NOAA-GMD) surface measurements (WMO, 2017).

Atmospheric N$_2$O is emitted from both natural (∼60%) and anthropogenic sources (∼40%), including oceans, soils, biomass burning, fertilizer use and various industrial processes (WMO, 2014). Among them, the increasing use of fertiliser is likely responsible for 80% of the increase in N$_2$O concentrations (Park et al., 2012). Global emissions of N$_2$O are difficult to estimate due to their heterogeneity in space and time.

Ground-based Fourier Transform Infrared (FTIR) spectrometers allow regular measurements of vertical total or partial col-

umn gas abundances in the atmosphere using solar absorption spectra. There are two well-known international networks based on ground-based solar FTIR instruments: the Total Carbon Column Observing Network (TCCON) established in 2004 (Wunch et al., 2011) and the Network for the Detection of Atmospheric Composition Change - the InfraRed Working Group (NDACC-IRWG; named NDACC in this study) established in 1991 (De Mazière et al., 2018). Both TCCON and NDACC networks have more than 20 sites around the world. TCCON and NDACC measurements can be made using the same instruments, with

different detectors and retrieval strategies. Some sites perform both TCCON and NDACC measurements simultaneously. N$_2$O is a target species of both networks. TCCON derives N$_2$O total columns from near-infrared (NIR) spectra recorded with an indium gallium arsenide (InGaAs) detector and NDACC derives N$_2$O total columns and vertical profiles from mid-infrared (MIR) spectra recorded with an indium antimonide (InSb) detector. NDACC N$_2$O total columns or vertical profiles have been used to study the long-term trend of N$_2$O (Zander et al., 1994; Angelbratt et al., 2011) and to evaluate MIPAS, ACE-FTS,

AIRS and IASI satellite measurements (Vigouroux et al., 2007; Strong et al., 2008; Xiong et al., 2014; García et al., 2016). TCCON dry-air total column-averaged abundance of N$_2$O (X$_{N_2O}$) measurements have been applied to assess the performance of an atmospheric general circulation model-based chemistry transport model (Saito et al., 2012).





Global chemical transport models (CTMs) are able to simulate the $N_2O$ concentration in the atmosphere. Prather et al. (2015) used four independent CTMs together with Microwave Limb Sounder (MLS) satellite measurements to estimate the lifetime of $N_2O$ in the atmosphere. Thompson et al. (2014) compared five CTM simulations with different atmospheric inversion frameworks. Large discrepancies existed for the regions of South and East Asia and for tropical and South America due to

the lack of observations from these places. Wells et al. (2015) described a 4D-Var inversion framework for $N_2O$ based on the GEOS-Chem CTM, and evaluated the utility of different observing networks for constraining $N_2O$ sources and sinks. Subsequently, Wells et al. (2018) applied the same model framework in a multi-inversion approach to place new top-down constraints on global $N_2O$ emissions.

To our knowledge, there have not yet been any studies investigating differences between the TCCON and NDACC $N_2O$

measurements. In this paper, an inter-comparison between the TCCON and NDACC $X_{N_2O}$ measurements at seven sites in the 2007-2017 period is carried out. The target of this study is that to better understand the discrepancies between the TCCON and NDACC $N_2O$ measurements, and to know whether two networks can be combined with atmospheric chemistry models for evaluation, seasonal cycles and long-term trend analyses. Sect. 2 describes the TCCON and NDACC data used in this paper. The biases between TCCON and NDACC $X_{N_2O}$ measurements are shown in Sect. 3. After that, discrepancies between the

two datasets at a high-latitude site are investigated in terms of their respective a priori profiles and vertical sensitivities. Next, $X_{N_2O}$ trends and seasonal cycles derived from the TCCON and NDACC and the nearby surface flask sample measurements are compared to the GEOS-Chem simulations in Sect. 5. Finally, conclusions are drawn in Sect. 6.

## 2 TCCON and NDACC measurements

The ground-based FTIR sites used in this study are shown in Figure 1. Both TCCON and NDACC $N_2O$ measurements are

available at these sites. The coordinates of the sites together with the time coverages of the data are listed in Table 1. Note that there are two observatories at Reunion Island, one is at St Denis recording NIR spectra and the other one is at Maïdo recording MIR spectra (Zhou et al., 2016). At Lauder, two spectrometers Bruker 120HR (2004 - 2011) and 125HR (2010 - present) have been applied to record TCCON spectra, and the same Bruker 120HR instrument is applied to record NDACC spectra. Details on the measurements can be found in Pollard et al. (2017). In this study, only the TCCON measurements from the Bruker

125HR at Lauder are used. At the other five sites, a single spectrometer measures for both networks.

The GGG2014 algorithm is applied to retrieve $X_{N_2O}$ from TCCON spectra, and it performs a profile scaling retrieval. $X_{N_2O}$ is obtained from the ratio between the total column of $N_2O$ ($TC_{N_2O}$) and $O_2$ ($TC_{O_2}$) (Yang et al., 2002)

$$X_{N_2O} = 0.2095 \times \frac{TC_{N_2O}}{TC_{O_2}} \frac{1}{\alpha \cdot [1 + \beta \cdot SBF(\theta)]}, \tag{1}$$

where 0.2095 is the constant volume mixing ratio (VMR) of the $O_2$ in the dry air; $\theta$ is the solar zenith angle (SZA); $SBF(\theta) =$

$[(\theta + 13)/(90 + 13)]^3 - [(45 + 13)/(90 + 13)]^3$; $\alpha$ is the scaling factor and $\beta \cdot SBF(\theta)$ is the empirically-derived airmass-dependent correction factor (Wunch et al., 2011, 2015). TCCON $X_{N_2O}$ measurements have been calibrated and validated with several HIPPO aircraft measurements over Wollongong (Australia), Lauder (New Zealand) and Four Corners (USA), and a



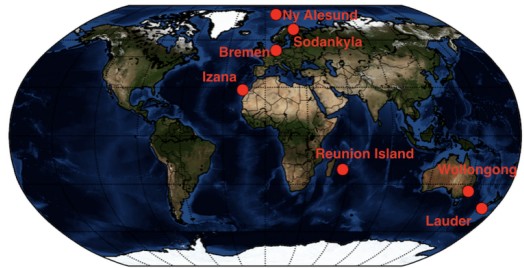

**Figure 1.** The location of the FTIR sites providing both TCCON and NDACC N$_2$O measurements used in this study.

**Table 1.** Characteristics of the FTIR sites contributing to the present work: location, altitude (in km a.s.l.), research team and time coverage of data. Note that there are two observatories at Reunion Island, one is at St Denis ('St') performing TCCON measurements and the other one is at Maïdo ('Ma') performing NDACC measurements.

| Site | Latitude | Longitude | Altitude (km a.s.l) | Team | Time coverage (TCCON/NDACC) | Instrument |
|------|----------|-----------|---------------------|------|------------------------------|------------|
| Ny-Ålesund | 78.9°N | 11.9°E | 0.02 | U. of Bremen | 2007-2017/2007-2017 | Bruker 120HR |
| Sodankylä | 67.4°N | 26.6°E | 0.19 | FMI & BIRA | 2009-2017/2012-2017 | Bruker 125HR |
| Bremen | 53.1°N | 8.8°E | 0.03 | U. of Bremen | 2009-2017/2007-2016 | Bruker 125HR |
| Izaña | 28.3°N | 16.5°W | 2.37 | AEMET & KIT | 2007-2017/2007-2017 | Bruker 125HR |
| Reunion Island | 21.0°S | 55.4°E | 0.08/2.16 (St/Ma) | BIRA | 2011-2017/2013-2017 | Bruker 125HR |
| Wollongong | 34.4°S | 150.9°E | 0.03 | U. of Wollongong | 2008-2017/2008-2017 | Bruker 125HR |
| Lauder | 45.0°S | 169.7°E | 0.37 | NIWA | 2010-2017/2007-2017 | Bruker 120/5HR |

START-08 measurement over Park Falls (USA). One calibration factor ($\alpha$) of 0.96 ($\pm0.01$) is applied to correct the systematic error in TCCON X$_{N_2O}$ data. Therefore, only a random uncertainty of about 1.0% is reported for TCCON data (Wunch et al., 2015). The a priori profile of TCCON is generated on a daily basis by a stand alone code (Toon and Wunch, 2014). The a priori VMR profiles of TCCON are based on MkIV balloon and ACE-FTS profiles measured in the 30-40°N latitude range from
5    2003 to 2007, which take into account the tropopause height variation and the secular trend.

NDACC uses either the SFIT4 algorithm (an updated version of SFIT2 (Pougatchev et al., 1995)) or the PROFFIT9 algorithm (Hase et al., 2004) to retrieve N$_2$O vertical profiles. Good agreement between these two retrieval algorithms has been demonstrated (Hase et al., 2004). Since the O$_2$ total column is not available from the MIR spectrum and the weak N$_2$ signal in the MIR region leads to a large scatter, the NDACC X$_{N_2O}$ is calculated from the dry-air column

$$X_{N_2O} = \frac{TC_{N_2O}}{P_s/(g \cdot m_{air}^{dry}) - TC_{H_2O}(m_{H_2O}/m_{air}^{dry})}, \tag{2}$$

where $TC_{H_2O}$ is total column of H$_2$O; $P_s$ is the surface pressure; g is the column-averaged gravitational acceleration; $m_{H_2O}$ and $m_{air}^{dry}$ are molecular masses of H$_2$O and dry air, respectively (Deutscher et al., 2010; Zhou et al., 2018). The total column



of $N_2O$ is calculated by integrating the partial column of each layer. For each site, the mean of the monthly means during 1980-2020 from the Whole Atmosphere Community Climate Model (WACCM) version 4 is applied to be the a priori profile for the NDACC retrievals (constant in time). There is no post-correction for NDACC retrievals. Therefore, the systematic uncertainty (about 2.0%) of NDACC $N_2O$ is reported together with the random uncertainty (about 1.5%), and the systematic uncertainty

of NDACC $N_2O$ total column is mainly due to uncertainties in the spectroscopic parameters (García et al., 2018).

The main differences between the TCCON and NDACC $X_{N_2O}$ retrieval strategies are listed in Table 2.

**Table 2.** The main differences between the TCCON and NDACC $X_{N_2O}$ measurements.

|  | TCCON | NDACC |
| --- | --- | --- |
| Retrieval algorithm | GGG2014 | SFIT4 or PROFFIT9 |
| Retrieval strategy | profile scaling | profile retrieval |
| Spectral range | NIR | MIR |
| A priori profile | GGG2014 code (daily) | WACCM v4 (fixed) |
| Airmass calculation | $O_2$ | surface pressure and $H_2O$ |
| Post-processing | calibrated by aircraft measurements | none |
| Systematic/random uncertainty | -/1.0% | 2.0/1.5% |

Both instrumental and retrieval settings for TCCON measurement are very consistent throughout the network (Wunch et al., 2011). The GGG2014 algorithm uses three retrieval windows (4373.5–4416.9 and 4418.55–4441.65; 4682.95–4756.05 cm$^{-1}$) and the atm.101 spectroscopy (Toon, 2014) to retrieve the total column of $N_2O$ (Notholt et al., 2014b; Kivi et al., 2014; Notholt

et al., 2014a; Blumenstock et al., 2014; De Mazière et al., 2014; Griffith et al., 2014; Sherlock et al., 2014). NDACC retrieval strategies can vary from site to site, depending on site-specific conditions, e.g. humidity, instrument and retrieval software. Table 3 lists the NDACC retrieval settings for each site. Two microwindows (2441.8–2444.6, 2481.1–2482.5 cm$^{-1}$) are employed at Ny-Ålesund and Bremen, while the other sites use four microwindows (2481.3-2482.6, 2526.4-2528.2, 2537.85-2538.8 and 2540.1-2540.7 cm$^{-1}$). The Wollongong site uses the atm.101 spectroscopy, while the other sites use the HITRAN2008 (Roth-

man et al., 2009). In fact, $N_2O$ line parameters are same in these two spectroscopy. The Optimal Estimation Method (OEM) (Rodgers, 2000) is applied to construct the regularization matrix of the a priori information at Ny-Ålesund, Bremen, Wollongong and Lauder, while the Tikhonov method (Tik) (Tikhonov, 1963) is applied at Sodankylä, Izaña and Reunion Island. The degrees of freedom for signal (DOFS) at these sites are in the range of 2.4–4.5. The range in DOFS is quite large; while it is know in the NDACC community that the DOFS of $N_2O$ retrieval is usually between 2.5-3.5 (Angelbratt et al., 2011; García

et al., 2018). The wide range of DOFS in this study does not affect the total column, but we limit to 3 partial columns for NDACC vertical profiles. To better understand the influence of these settings, we compare the mean and standard deviation (std) of one-year NDACC retrieved $X_{N_2O}$ in 2014 at Reunion Island after changing the spectroscopy, regularization method, or retrieval windows (see Table 4). There is no difference after changing the spectroscopy from the HITRAN2008 to the atm.101. Changing the regularization method from OEM to Tik introduces a difference of 0.28 ppb or 0.09% which is negligible com-





pared to the reported uncertainty. The maximum difference (0.78 ppb or 0.25%) occurs after changing the retrieval windows from 4 to 2 microwindows. The systematic and random uncertainties of the NDACC $N_2O$ retrievals are about 2.0 and 1.5 %, respectively. Since the difference in Table 4 is within the retrieval uncertainties of TCCON and NDACC, and there is no seasonal variation in the difference. Consequently, it is assumed that the influences caused by these retrieval settings can be

5    ignored.

**Table 3.** NDACC retrieval settings at seven FTIR sites. For sites using two microwindows, retrieval windows are 2441.8-2444.6 and 2481.1-2482.5 cm$^{-1}$. For sites using four microwindows, retrieval windows are 2481.3-2482.6, 2526.4-2528.2, 2537.85-2538.8 and 2540.1-2540.7 cm$^{-1}$.

| Site | Code | Spectroscopy | Regularization | Retrieval windows | DOFS (mean ± std) |
|------|------|--------------|----------------|-------------------|-------------------|
| Ny-Ålesund | SFIT4 | HITRAN2008 | OEM | 2 MWs | 3.9±0.2 |
| Sodankylä | SFIT4 | HITRAN2008 | Tik | 4 MWs | 2.4±0.1 |
| Bremen | SFIT4 | HITRAN2008 | OEM | 2 MWs | 4.5±0.3 |
| Izaña | PROFFIT9 | HITRAN2008 | Tik | 4 MWs | 2.9±0.2 |
| Reunion Island | SFIT4 | HITRAN2008 | Tik | 4 MWs | 2.9±0.2 |
| Wollongong | SFIT4 | atm.101 | OEM | 4 MWs | 3.8±0.2 |
| Lauder | SFIT4 | HITRAN2008 | OEM | 4 MWs | 3.4±0.2 |

**Table 4.** NDACC retrieved $X_{N_2O}$ in 2014 with different settings (spectroscopy + regularization + retrieval windows + a priori profile) at Reunion Island.

| settings | $X_{N_2O}$ (mean ± std [ppb]) |
|----------|------------------------------|
| HITRAN2008+Tik+4MWs+WACCM | 312.63 ± 1.16 |
| atm.101+Tik+4MWs+WACCM | 312.63 ± 1.16 |
| HITRAN2008+OEM+4MWs+WACCM | 312.35 ± 1.28 |
| HITRAN2008+Tik+2MWs+WACCM | 311.85 ± 1.35 |
| HITRAN2008+Tik+4MWs+TCCONap | 312.44 ± 1.22 |

The retrieved FTIR (TCCON and NDACC) $N_2O$ total column relates to the true state of the atmosphere and the a priori information via (Rodgers, 2003)

$$TC_r = TC_a + \boldsymbol{A} \cdot (\boldsymbol{PC_t} - \boldsymbol{PC_a}) + \varepsilon, \tag{3}$$

where $TC_r$ and $TC_a$ are the retrieved and a priori $N_2O$ total columns respectively; $\boldsymbol{PC_a}$ and $\boldsymbol{PC_t}$ are the a priori and the true

10    $N_2O$ partial column profiles respectively; $\boldsymbol{A}$ is the column averaging kernels (AVK) of the TCCON and NDACC retrievals, representing the vertical sensitivity of the retrieved $N_2O$ to the true state; $\varepsilon$ is the error. Figure 2 shows the TCCON and NDACC averaging kernels. Whereas NDACC exhibits uniform sensitivity throughout the troposphere and stratosphere, the




TCCON sensitivity increases with altitude. As a result, TCCON retrievals will tend to underestimate a deviation from the a priori in the lower troposphere, and overestimate it in the stratosphere. We also test the NDACC retrievals by using the TCCON a priori profile as the a priori profile at Reunion Island (see the last row in Table 4). The difference between the results using the WACCM model and the TCCON code as the a priori is relatively small (0.19 ppb or 0.06%), because the AVK of NDACC

is very close to 1.0. It is thus assumed that the NDACC retrieved $N_2O$ total column is independent of the a priori profile. According to Rodgers (2003), the difference between retrieved $N_2O$ total column from TCCON and NDACC can be written as

$$TC_{N_2O,ndacc} - TC_{N_2O,tccon} = (\boldsymbol{A_{ndacc}} - \boldsymbol{A_{tccon}})(\boldsymbol{PC_t} - \boldsymbol{PC_{tcconap}}). \tag{4}$$

Therefore, apart from the spectroscopy causing a bias between different retrieval windows, the difference between retrieved

$N_2O$ total column from TCCON and NDACC is mainly due to their AVK differences, and the difference in the $N_2O$ partial column profile between the TCCON a priori and the true state.

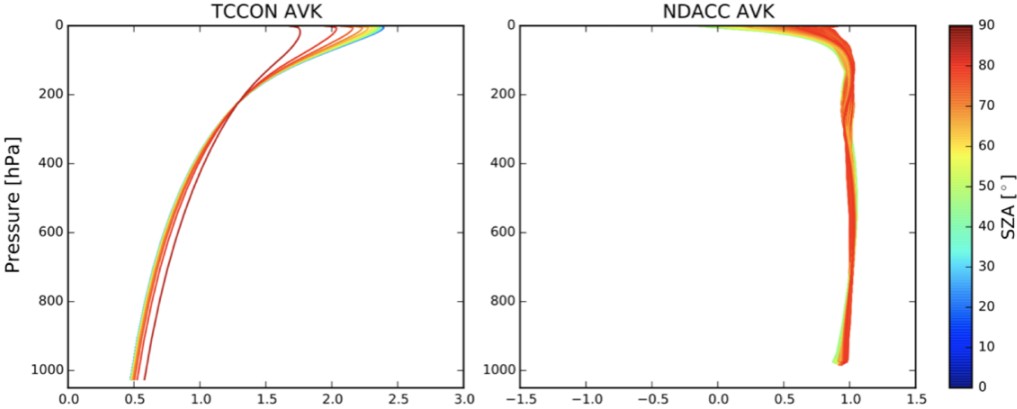

**Figure 2.** The typical $N_2O$ column averaging kernel of TCCON (left panel) and NDACC (right panel) at Reunion Island. The different colors correspond to different SZAs.

## 3   Comparison between TCCON and NDACC $X_{N_2O}$ measurements

The time series of TCCON and NDACC $X_{N_2O}$ measurements together with their differences are shown in Figure 3. The statistical results of the co-located hourly means of TCCON and NDACC measurements are listed in Table 5. Note that the

NDACC $X_{N_2O}$ at Reunion Island is multiplied with a factor of 1.006 to correct the surface altitude difference between St Denis (85 m a.s.l.) and Maïdo (2155 m a.s.l.). The factor of 1.006 is calculated from the ratio of the 0.085 – 100 km $N_2O$ partial column to the 2.155 – 100 km partial column based on the WACCM v4 model.

The averaged biases between the NDACC and TCCON $X_{N_2O}$ measurements (NDACC-TCCON) at these sites are from -3.32 to 1.37 ppb (-1.1 – 0.5 %) with the standard deviations of 1.69 – 5.01 ppb (0.5 – 1.6 %). Since the random uncertainty of the



TCCON measurement is about 1.0% and the systematic and random uncertainties of the NDACC $N_2O$ retrievals are about 2.0 and 1.5 %, the difference between the TCCON and NDACC measurements are within their combined uncertainty. However, there is a large difference between TCCON and NDACC data in February-May at Ny-Ålesund and Sodankylä, which will be discussed in the next section. In addition, the Fig. 3 shows that the bias between the NDACC and TCCON measurements

5   increases with time, because the $X_{N_2O}$ trend derived from NDACC measurements is slightly larger than that derived from TCCON measurements. The reason will be explained in Sect. 5.

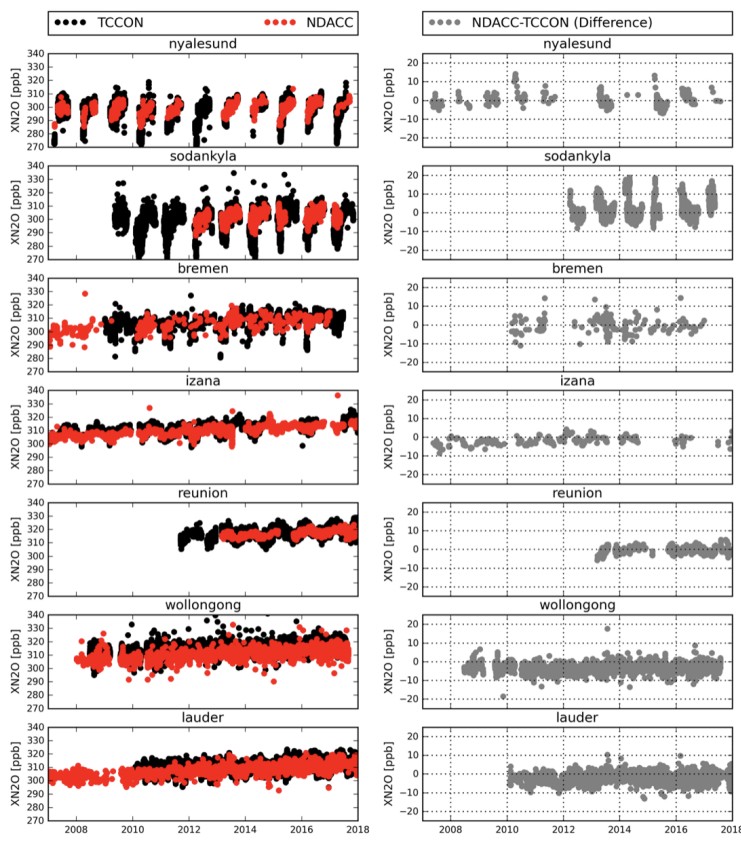

**Figure 3.** Time series of TCCON and NDACC retrieved $X_{N_2O}$ (left panels) together with the differences (NDACC minus TCCON) between their co-located hourly means (right panels) at seven sites.

## 4   Case study - Sodankylä

The time series of TCCON and NDACC co-located $X_{N_2O}$ hourly means together with their difference and correlation at Sodankylä are shown in Figure 4. There is no FTIR measurement during the northern winter season due to the polar night. The

10   TCCON $X_{N_2O}$ measurements are very close to the NDACC data in northern summer and autemn seasons, but are lower than





**Table 5.** The mean and the standard deviation (std) of the difference between co-located hourly means of TCCON and NDACC data, together with the correlation coefficient (R) and total number (N) of the co-located data pairs.

| Site | mean [ppb] | std [ppb] | R | N |
|---|---|---|---|---|
| Ny-Ålesund | 0.43 | 4.23 | 0.82 | 326 |
| Sodankylä | 1.37 | 5.01 | 0.87 | 2498 |
| Bremen | -0.24 | 4.21 | 0.67 | 167 |
| Izaña | -1.85 | 2.04 | 0.78 | 232 |
| Reunion Island | 1.02 | 1.69 | 0.81 | 619 |
| Wollongong | -3.32 | 2.13 | 0.78 | 4906 |
| Lauder | -1.96 | 2.60 | 0.69 | 2331 |

the NDACC data during spring. The air above Sodankylä is frequently affected by the Arctic polar vortex in winter and spring (Kivi et al., 2001, 2007; Karppinen et al., 2016; Denton et al., 2018). The high potential vorticity (PV) value on a constant potential temperature of 430 K is a useful index to identify polar vortex (Schoeberl and Hartmann, 1991). The PV data in this study is downloaded from the ECMWF ERA-Interim reanalysis dataset (Dee et al., 2011). We find that the low $X_{N_2O}$ values

in the TCCON measurements in Figure 4 correspond to periods of high PV, indicating that Sodankylä is inside polar vortex. During that time, stratospheric composition is controlled by a large mass of cold and dense Arctic air. $N_2O$ decreases rapidly above the tropopause due to chemical conversion to NO globally. However, in the arctic winter the air descends due to the denser cold air in the polar night and the isolation from mid-latitude refreshing. As the $N_2O$ VMR decreases with altitude during subsidence, the VMR at each altitude is less and the total column decreases. Similar issue has been found by Ostler

et al. (2014) for the TCCON $X_{CH_4}$ measurements at Ny-Ålesund influenced by polar vortex subsidence.

$N_2O$ measurements from the Atmospheric Chemistry Experiment–Fourier Transform Spectrometer (ACE-FTS) satellite are applied to assess the change of the $N_2O$ vertical profile when Sodankylä is inside polar vortex. ACE-FTS uses the solar occultation technique to measure the mole fractions of atmospheric trace gases, mainly in the stratosphere, with a vertical resolution between 1.5 and 6 km (Boone et al., 2013). The latest ACE-FTS level 2 v3p6 $N_2O$ data is used in this study. It is

assumed that ACE-FTS measurements are representative of the $N_2O$ variablity in the stratosphere. Sheese et al. (2017) showed that the differences between ACE-FTS v3p6 and MLS and MIPAS $N_2O$ measurements are within 20% below 45 km. ACE-FTS pixels are selected within $\pm 4 \times 8°$ (latitude by longitude) of Sodankylä during 2012-2016. In total, there are 43 individual days when TCCON, NDACC and ACE-FTS measurements are all available. The day is identified as being within polar vortex if it satisfies the following two criteria: 1) PV value at 430K on that day is larger than $20 \times 10^{-6} Km^2 kg^{-1} s^{-1}$; 2) the daily mean

of $X_{N_2O}$ derived from TCCON differs by more than 6.0 ppb from the corresponding daily mean of NDACC data. The second criteria is added to avoid the days when the polar vortex just starts or ends, while the TCCON and NDACC spectra are recorded on the same day but outside the polar vortex system. As a result, 3 (25 March 2015, 16 February 2016 and 24 March 2016) out of these 43 days are identified as inside polar vortex. Figure 5 shows the NDACC a priori profile, TCCON a priori profile,



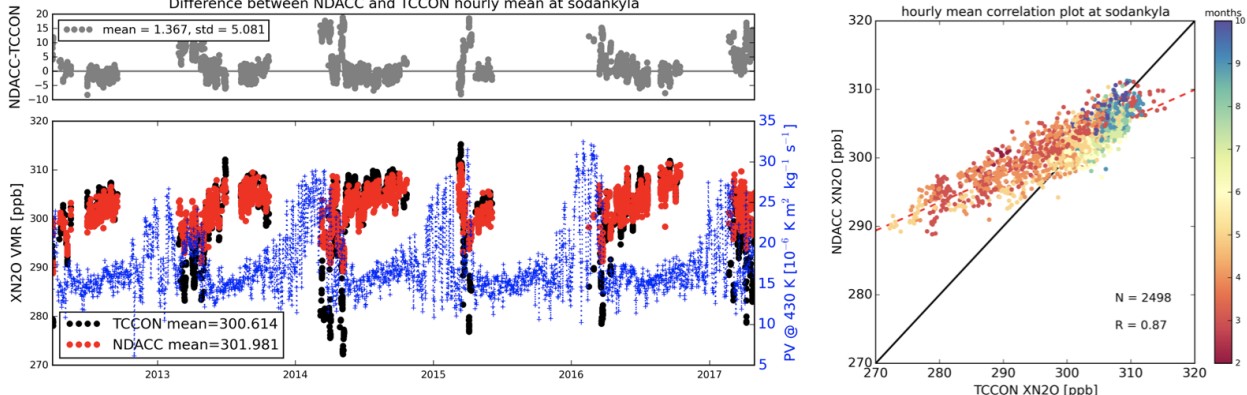

**Figure 4.** The time series of the hourly means from the TCCON and NDACC $X_{N_2O}$ measurements at Sodankylä, together with the absolute difference (unit: ppb) between them (left lower and top, respectively) and their correlation (right panel) colored according to the measurement month (right). Along with the $X_{N_2O}$ measurements, the blue line in the left bottom panel is the potential vorticity (PV) value on a constant potential temperature of 430 K above Sodankylä.

NDACC retrievals, collocated ACE-FTS measurements and the ACE-FTS measurements smoothed with the NDACC a priori profile and AVK on inside-vortex (3) and outside-vortex (40) days. It is confirmed by the ACE-FTS measurements that the $N_2O$ VMR rapidly decreases more rapidly above the tropopause height when polar vortex occurs. The smoothed ACE-FTS measurements are close to the NDACC retrieved $N_2O$ profiles for both inside and outside polar vortex cases, because the

NDACC retrieval has a good sensitivity and the NDACC retrieval is able to capture the change in the stratosphere. However, the TCCON retrieval overestimates the deviation from the a priori in the stratosphere (see Figure 2). When Sodankylä is inside polar vortex, the ACE-FTS measurement (used here as reference dataset) is much lower than the TCCON a priori profile in the stratosphere. As a result, the TCCON retrieved $N_2O$ column overestimates the magnitude of the $N_2O$ decrease, and explaining why these data are always lower than the NDACC measurements in spring during polar vortex overpasses.

Figure 6 compares the standard TCCON and NDACC $X_{N_2O}$ retrievals with updated versions using the ACE-FTS measurement as a priori profile (above 10 km) for days inside polar vortex. As expected, changing the a priori profile does not lead to much change in the NDACC retrievals, whereas the TCCON retrievals using the ACE-FTS profile as a priori profile increase significantly and are more similar to the NDACC retrievals. After updating the a priori profile, the mean difference in $X_{N_2O}$ between TCCON and NDACC at these 3 days reduces from 11.5 ppb to 1.2 ppb. Based on this experiment, the averaged $N_2O$

profile from the ACE-FTS measurements on these 3 days is applied to be a priori profile for all the TCCON retrievals inside polar vortex. The time series of the updated TCCON and original NDACC retrievals and their correlation plot are shown in Figure 7. The discrepancy between TCCON and NDACC $X_{N_2O}$ measurements in spring is almost eliminated. The mean and standard deviation of the difference between TCCON and NDACC $X_{N_2O}$ decrease to -0.74 and 2.81 ppb. The R values between





TCCON and NDACC $X_{N_2O}$ measurements are very similar in Figures 4b and 7b, but in Figure 7b the slope of the regression line increases from 0.41 to 0.63 along with a smaller y-intercept value.

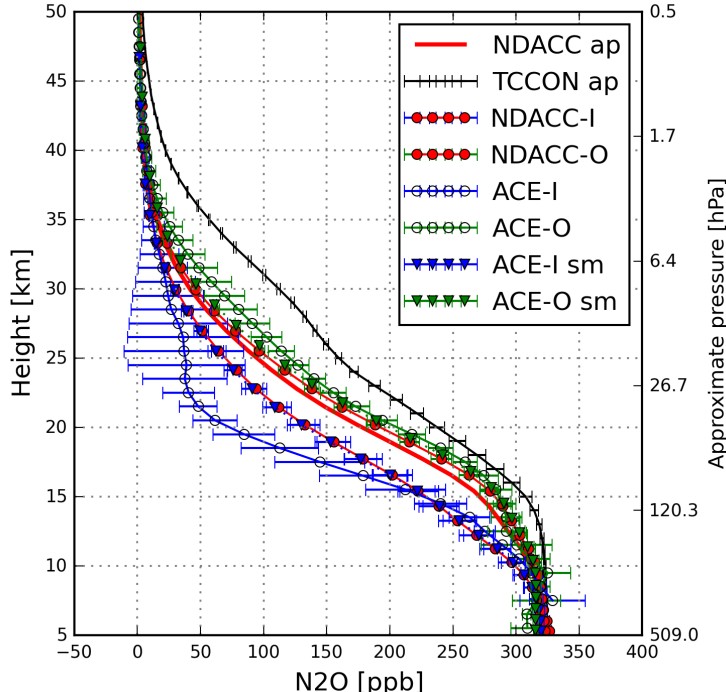

**Figure 5.** N$_2$O profiles from the NDACC a priori profile (NDACC ap), TCCON a priori profile (TCCON ap), NDACC retrievals inside/outside polar vortex (NDACC-I/NDACC-O), co-located ACE-FTS measurements inside/outside polar vortex (ACE-I/ACE-O) and the ACE-FTS measurements smoothed with the NDACC AVK inside/outside polar vortex (ACE-I sm/ACE-O sm).

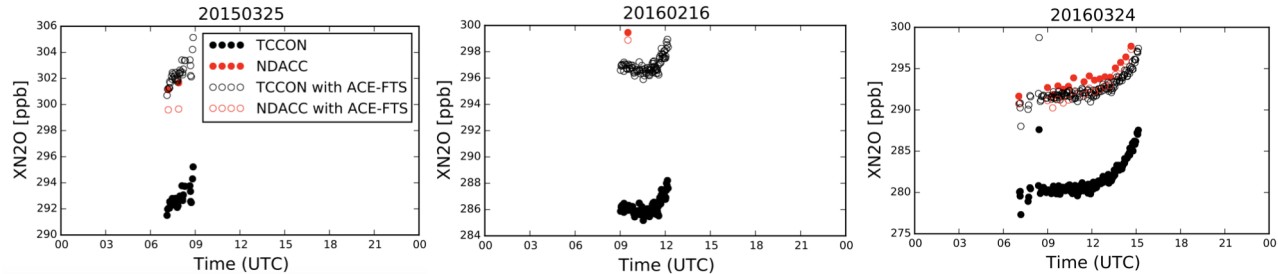

**Figure 6.** The standard TCCON and NDACC retrieved $X_{N_2O}$ and updated retrieved $X_{N_2O}$ using the ACE-FTS measurement as the a priori profile in the stratosphere on days when Sodankylä is inside polar vortex (25 March 2015, 16 February 2016 and 24 March 2016).





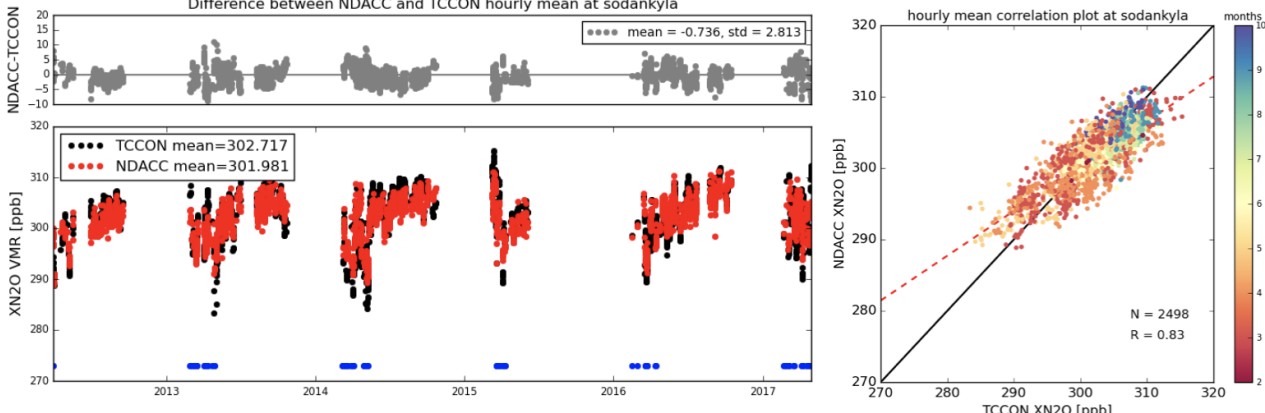

**Figure 7.** Same as Figure 4, but showing the TCCON retrievals on the days inside polar vortex (blue dots) using the ACE-FTS measurement as the a priori profile.

## 5    Comparison between FTIR measurements with GEOS-Chem model

### 5.1    GEOS-Chem model simulation

Here we compare the TCCON and NDACC measurements with simulated $N_2O$ fields from the GEOS-Chem CTM to better understand trends and seasonal cycles in atmospheric $N_2O$. The GEOS-Chem simulations shown here, described in detail by Wells et al. (2015, 2018), are driven by MERRA-2 reanalysis data. The a priori simulation uses $N_2O$ emissions from the O-CNv1.1 land surface model (Zaehle et al., 2011) for soils, the marine biogeochemistry model PlankTOM5 (Buitenhuis et al., 2010) for ocean, the Emission Database for Global Atmospheric Research EDGARv4.2 FT2010 (European Commission, 2013) for non-soil anthropogenic sources, and the Global Fire Emission Database GFEDv4.1s (Van Der Werf et al., 2017) for biomass burning. These a priori inventories correspond to a global flux of 17.9-18.8 TgN/year for 2007-2014. In the a posteriori simulation, $N_2O$ surface fluxes in the model have been optimized on the basis of surface measurements using a 4D-Var inversion framework as described by Wells et al. (2018). The a posteriori global flux ranges from 15.5-17.9 TgN/year. Stratospheric loss of $N_2O$ by photolysis and reaction with $O(^1D)$ is included in the model and leads to an atmospheric lifetime of approximately 127 years.

Global GEOS-Chem output shown here are monthly averages for 2007-2014, with horizontal resolution of 4° latitude × 5° longitude and 47 vertical levels from the surface to 0.01 hPa. Model grid points closest to the FTIR stations are employed for comparison with the TCCON and NDACC data. Following Eq. 2, the column-averaged $N_2O$ from the model a priori and a posteriori simulations are derived to compare with TCCON and NDACC measurements.



## 5.2 Computation method for trend and seasonal variation

As atmospheric $N_2O$ has been continuously increasing over the past decade (WMO, 2017), a linear regression model is used to calculate the $N_2O$ trend.

$$Y(t) = A_0 + A_1 \cdot t + \sum_{k=1}^{3}(A_{2k}\cos(2k\pi t) + A_{2k+1}\sin(2k\pi t)) + \varepsilon(t), \tag{5}$$

where $Y(t)$ is measured or modeled $N_2O$; $A_1$ is the $N_2O$ trend, and $A_2 - A_7$ are the amplitudes of the periodic variations during the year. Then, the detrended data $(Y(t)_d)$ is calculated as

$$Y(t)_d = Y(t) - (A_0 + A_1 \cdot t). \tag{6}$$

The seasonal variation is represented by the monthly means of the detrended data and their associated uncertainty ($2\,\sigma$).

## 5.3 N₂O trends

The calibrated $N_2O$ measurements from weekly surface air samples collected in glass flasks during 2007-2014 from the Earth System Research Laboratory NOAA-GMD are used as a reference to compare with FTIR measurements and the model simulation. Uncertainties of the surface measurements are about 0.3 ppb (Dlugokencky et al., 2018). As most FTIR sites are not installed with a flask sampling system, we use the closest sampling site within 1000 km of each FTIR site to compare with TCCON and NDACC measurements and model output. Note that there is no flask sampling system available near Reunion

Island. Table 6 lists the GMD sites used in this study and their corresponding TCCON and NDACC sites.

**Table 6.** Locations of the flask sampling data around each FTIR site. There is no flask sampling site available near Reunion Island.

| NOAA-GMD site | lat/lon | altitude (km a.s.l.) | FTIR site |
|---|---|---|---|
| Ny-Ålesund, Svalbard (ZEP) | 78.9N/11.9E | 0.47 | Ny-Ålesund |
| Pallas-Sammaltunturi (PAL) | 70.0N/24.1E | 0.56 | Sodankylä |
| Ochsenkopf (OXK) | 50.0N/11.8E | 1.02 | Bremen |
| Izaña (IZO) | 28.3N/16.5W | 2.37 | Izaña |
| Cape Grim (CGO) | 40.7S/144.7E | 0.09 | Wollongong |
| Baring Head (BHD) | 41.4S/174.9E | 0.08 | Lauder |

Figure 8 shows the $X_{N_2O}$ trends from flask sample measurements, TCCON and NDACC FTIR retrievals, and the a priori and a posteriori model simulations at each site. Note that model output and flask sample data are both for the 2007-2014 period, whereas all available FTIR measurements during the 2007-2017 period (see Fig. 3). The numbers of FTIR measurements before 2014 are very limited at Sodankylä and Reunion Island. As the NOAA-GMD surface $N_2O$ measurements show that

atmospheric $N_2O$ increases with a constant annual growth rate during the last decade, it is assumed that these two different time periods do not introduce the discrepancy in the trend and seasonal cycle computations. The a priori GEOS-Chem $X_{N_2O}$





trend (about 1.25 ppb/year) is too large based on all the observational datasets in Figure 8, implying an $N_2O$ flux overestimate in the a priori inventories used in the model. On the other hand, the $X_{N_2O}$ trend in the a posteriori GEOS-Chem simulation is very close to that seen in the NDACC and surface datasets, except at Ny-Ålesund.

The $X_{N_2O}$ trend derived from TCCON measurements (0.8 – 0.9 ppb/year) is slightly smaller compared to the results from

NDACC and flask sample measurements (0.9 – 1.0 ppb/year). The TCCON AVK (Figure 2) indicates that the TCCON retrieval in the lower and middle troposphere includes a 30-50% contribution from the a priori assumption (Eq. 3). As mentioned in Sect. 2, TCCON uses a stand alone code to create the a priori profile for each site (Toon and Wunch, 2014). The a priori $N_2O$ has a trend of 0.1 %/year, which is much lower than the true state of the atmosphere (about 0.3 %/year) (WMO, 2017). Therefore, we update the TCCON retrieval using an new a priori $N_2O$ profile with an annual growth of 0.3 %/year, and keep

the $N_2O$ mole fraction on the first day of 2007 unchanged. After updating the a priori $N_2O$, the $X_{N_2O}$ trends from the TCCON measurements increase by 0.05-0.10 ppb/year at these sites, and the results are similar to the ones from the NDACC and the flask sample measurements.

Large uncertainties are found for the FTIR-based $X_{N_2O}$ trends at Ny-Ålesund and Sodankylä, because of a strong seasonal cycle in $X_{N_2O}$ at high latitude, the intensity of the polar vortex varying from year to year, and gaps due to polar night. The

$X_{N_2O}$ trends from the TCCON and NDACC measurements at Ny-Ålesund are much smaller than the trends from the GEOS-Chem a posteriori simulation and flask sample measurements. This might be explained by the lack of measurements; the fact that no observations are possible during winter (absence of sun) and only limited measurements are available during the other seasons. For instance, there are no full extent of the minimum from NDACC $X_{N_2O}$ measurements at Ny-Ålesund in 2007, 2009 and 2011 compared to other years. The $X_{N_2O}$ trends from TCCON and NDACC measurements at Sodankylä are closer

to the results from the model simulations and in situ measurements, which is probably due to comparatively more FTIR spectra recorded at Sodankylä.

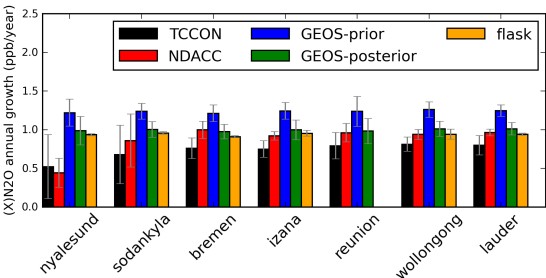

**Figure 8.** The $X_{N_2O}$ trends from TCCON and NDACC FTIR measurements (all available data during 2007-2017; see Figure 3), a priori and a posteriori model simulations (2007-2014), and surface $N_2O$ trend from flask sample measurements (2007-2014), together with their uncertainties at each site.





### 5.4  N$_2$O seasonal variations

The seasonal variations in X$_{N_2O}$ from the TCCON and NDACC measurements, a priori and a posteriori GEOS-Chem model simulations are shown in Figure 9. The seasonal variations of X$_{N_2O}$ from a priori and a posteriori GEOS-Chem model simulations are very similar. For the Ny-Ålesund and Sodankylä sites (high latitude in the Northern Hemisphere), FTIR measurements

and model simulations show a maximum during August-October and a minimum during February-April. The amplitude of the seasonal cycle seen in the NDACC measurements is slightly larger than that in the model simulation. The amplitude of the seasonal variations from TCCON measurements are much larger than that from NDACC measurements, because the TCCON measurements overestimate the contribution from the stratosphere and the stratospheric N$_2$O VMR is quite variable in the high latitude. For the Bremen and Izaña sites (middle latitude in the Northern Hemisphere), the seasonal variations from TC-

CON and NDACC measurements are in good agreement with those from the model simulation. X$_{N_2O}$ exhibits a maximum in August-October and a minimum in February-April. For Reunion Island, Wollongong and Lauder (low and middle latitude in the Southern Hemisphere), the seasonal X$_{N_2O}$ variations in the model simulations exhibit a maximum in August-October and a minimum in February-April, whereas the TCCON and NDACC measurements show the opposite pattern. The phases of the X$_{N_2O}$ seasonal cycles from the FTIR measurements are close to the model simulations in the Northern Hemisphere, while

large discrepancies are apparent in the Southern Hemisphere.

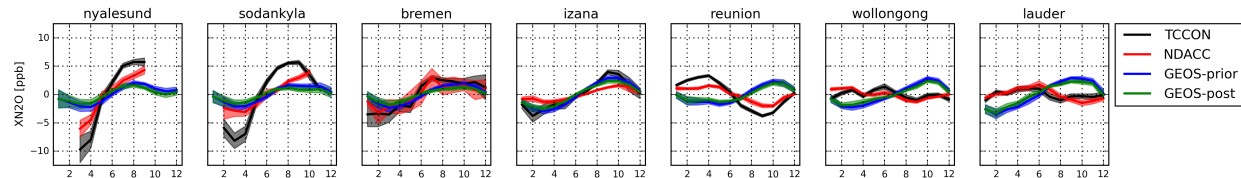

**Figure 9.** The X$_{N_2O}$ seasonal variations from TCCON and NDACC FTIR measurements (all available data during 2007-2017; see Figure 3), a priori and a posteriori model simulations (2007-2014) together with their uncertainties at each site.

Thompson et al. (2014) pointed out that many CTMs do not represent the seasonal cycle of Southern Hemisphere N$_2$O well, due to the lack of observations to constrain atmospheric inversions. The discrepancy in the seasonal cycle of Southern Hemisphere N$_2$O seen above could arise from a model misrepresentation of the stratosphere-troposphere exchange, errors in the seasonality of Southern Hemisphere emissions, or incorrect model transport of N$_2$O from lower latitudes. As the NDACC

measurements provide N$_2$O profiles with about 3 distinct partial columns (DOFS about 3.0; see Table 3), the model simulations are compared with NDACC measurements in three vertical ranges (surface–8, 8–17 and 17–50 km; each partial column has about 1.0 DOFS). In addition, surface flask sample measurements are employed to show the seasonal cycle of N$_2$O at the surface.

Figure 10 shows the N$_2$O seasonal variations from flask sample measurements and a priori and a posteriori model simulations

at the surface, and X$_{N_2O}$ seasonal variations from NDACC measurements and model simulations for three vertical ranges at Izaña, Reunion Island, Wollongong and Lauder. We mainly focus on the sites in the Southern Hemisphere, and Izaña is added to





represent a site in the Northern Hemisphere. The model a posteriori N$_2$O seasonal cycle at the surface is in a good agreement with that based on flask sample measurements at Izaña, but not at Wollongong and Lauder, which is consistent with the conclusions of Thompson et al. (2014) that lack of observations limit the accuracy of inversions in the Southern Hemisphere. However, below 8 km there is no clear seasonal cycles from NDACC measurements and GEOS-Chem a posteriori simulations,

and the uncertainties are about as large as the seasonal cycle amplitude. For the second layer (8-17 km), discrepancies between the NDACC measurements and the model simulations clearly exist at Wollongong and Lauder. According to the NCEP re-analysis data, the tropopause height at Izaña and Reunion Island is about 15–17 km, which is higher than that at Wollongong and Lauder (approximately 10–12 km). Therefore, the vertical range of 8–17 km is still in middle and upper troposphere for Izaña and Reunion Island, but is already in upper troposphere and lower stratosphere for Wollongong and Lauder. The seasonal

cycles of X$_{N_2O}$ between the model simulations and NDACC measurements are still in agreement at Izaña, but not at the sites in the Southern Hemisphere. The vertical range of 17–50 km is in the stratosphere for all sites. It is inferred that the X$_{N_2O}$ seasonal cycle discrepancy between model simulations and FTIR measurements in the Southern Hemisphere is dominated by their difference in the stratosphere, which is probably due to the misrepresentation of the stratosphere-troposphere exchange or the inappropriate N$_2$O transport or loss in the stratosphere. Further investigations are needed to understand why this discrepancy

occurs in the stratosphere in the Southern Hemisphere.

## 6 Conclusions

N$_2$O is an important greenhouse gas and it can generate nitric oxide, which depletes ozone in the stratosphere. It is a common target gas for both TCCON and NDACC networks. However, to our knowledge, no inter-comparison between both datasets is available in literature. In this study, a global view of the X$_{N_2O}$ measurement differences between these two networks is

presented at seven sites (Ny-Ålesund, Sodankylä, Bremen, Izaña, Reunion Island, Wollongong and Lauder). The mean and standard deviation of the difference between the NDACC and TCCON X$_{N_2O}$ (NDACC-TCCON) are -3.32 – 1.37 ppb (-1.1 – 0.5 %) and 1.69 – 5.01 ppb (0.5 – 1.6 %), which are within the uncertainties of the two datasets. The NDACC retrieval has good sensitivity throughout the troposphere and stratosphere, and the choice of the a priori profile has limited influence (within 0.1% for retrieved N$_2$O total column). The TCCON retrieval underestimates a deviation from the a priori in the troposphere

and overestimates it in the stratosphere. As a result, the TCCON X$_{N_2O}$ measurement is strongly affected by its a priori profile. The difference between TCCON and NDACC retrieved N$_2$O total columns is then mainly due to the AVK differences, and to N$_2$O profile differences between the TCCON a priori and the true state of the atmosphere. The case study at Sodankylä shows that TCCON X$_{N_2O}$ measurements are strongly affected by polar vortex. When Sodankylä is inside polar vortex, the N$_2$O VMR observed by the ACE-FTS satellite is much lower than the TCCON a priori value in the stratosphere. The TCCON retrieved

X$_{N_2O}$ is then much lower than the true state of the atmosphere because the TCCON retrieval overestimates a deviation from the a priori at high altitudes. This is the reason why TCCON measurements are always lower than NDACC measurements at high latitudes in spring during polar vortex overpasses.





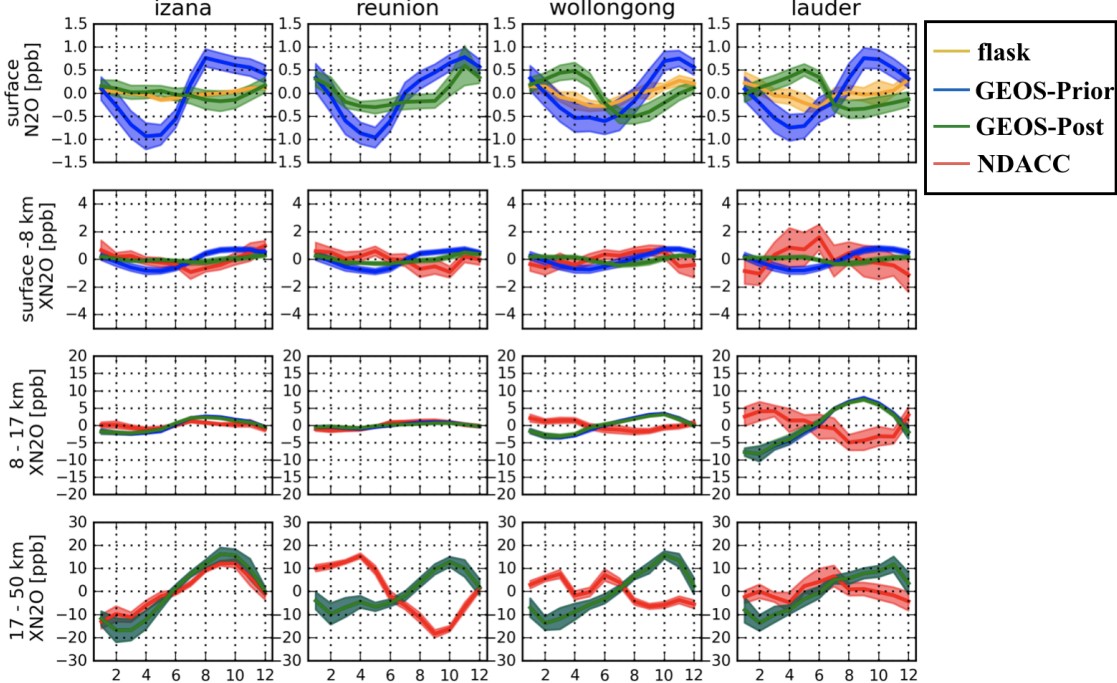

**Figure 10.** The $N_2O$ seasonal variations from the flask sample measurements and the prior and a posteriori model simulations at the surface at Izaña, Reunion Island, Wollongong and Lauder (top panels). Second to fourth panels show the $X_{N_2O}$ seasonal variations from NDACC measurements and the GEOS-Chem model simulations for three vertical ranges: surface–8 km, 8–17 km and 17–50 km. Note that the $X_{N_2O}$ seasonal variations from the GEOS-Chem a priori and a posterior simulations are almost same for the high altitude layers (8–17 km and 17–50 km).

Trends and seasonal cycles of $X_{N_2O}$ derived from TCCON and NDACC measurements, and nearby surface flask sample measurements are compared to the GEOS-Chem model a priori and a posteriori simulation. The a posteriori $N_2O$ fluxes are optimized based on surface $N_2O$ measurements within a 4D-Var inversion framework. The $X_{N_2O}$ trends from the GEOS-Chem a posteriori simulation are very close to those seen in the NDACC and flask sample measurements (0.9 – 1.0 ppb/year).

5     It is confirmed by the FTIR measurements that the $N_2O$ fluxes of the a priori inventories in the GEOS-Chem model are overestimated. The $X_{N_2O}$ trends of 0.8 – 0.9 ppb/year from TCCON measurements are slightly lower compared to the NDACC and flask sample measurements, because TCCON measurements have a 30-50 % contribution from the a priori in the lower and middle troposphere and the annual growth in the TCCON a priori (0.1%) is lower than the observed surface $N_2O$ concentration (0.3%). The seasonal variations of $X_{N_2O}$ from the GEOS-Chem model simulations are consistent with those from TCCON and

10     NDACC measurements in the Northern Hemisphere, but not in the Southern Hemisphere. A discrepancy exists between the surface samplings and the model a posteriori simulation in the Southern Hemisphere, and it is inferred that lack of observations limits the improvement in the $N_2O$ a posteriori fluxes. As NDACC measurements provide $N_2O$ profiles with about 3 distinct





partial columns, the model simulations are compared with NDACC measurements in three vertical ranges (surface–8, 8–17 and 17–50 km). It is found that the discrepancy in the $X_{N_2O}$ seasonal cycle between model simulations and FTIR measurements in the Southern Hemisphere is mainly due to stratospheric effects.

In summary, the TCCON and NDACC $X_{N_2O}$ measurements are in good agreement, and their differences are within the

combined uncertainty. However, due to the averaging kernels, TCCON $X_{N_2O}$ retrievals are strongly affected by a priori profiles while NDACC $X_{N_2O}$ retrievals can capture the tropospheric and stratospheric variations of $N_2O$ as well as the $X_{N_2O}$ trend very well using a fixed a priori profile. $X_{N_2O}$ trends from TCCON measurements are slightly underestimated because of the weak trend in its a priori. Fortunately, the issues of TCCON $X_{N_2O}$ measurements could be solved with an improved a priori.

*Data availability.* The TCCON data are publicly available through the TCCON wiki (https://tccondata.org/). The NDACC data except So-

dankylä are publicly available from the NDACC database (http://www.ndacc.org). The ACE-FTS data used are available from http://ace.uwaterloo.ca/data/ (registration required). The NOAA are available from the NOAA FTP server ftp://aftp.cmdl.noaa.gov/data/greenhouse_gases/n2o/flask/. The Sodankylä MIR data and the GEOS-Chem model data can be obtained by contacting the authors.

*Competing interests.* The authors declare that they have no conflict of interest.

*Acknowledgements.* Minqiang Zhou is supported by the Belgian Complementary Researchers program. We would like to thank TCCON

and NDACC networks for making the data publicly available. The FTIR sites at Reunion Island are operated by the BIRA-IASB and locally supported by LACy/UMR8105, Université de La Réunion. We would like to thank Nicolas Kumps, Bart Dils and Francis Scolas (BIRA-IASB) for their contributions to the FTIR measurements maintenance, and Edward Dlugokencky (NOAA) for sharing the flask sample measurements. Development of the GEOS-Chem N2O simulation was supported by NOAA (Grant #NA13OAR4310086) and the Minnesota Supercomputing Institute. The Lauder FTIR measurements are core funded by NIWA from New Zealand's ministry of business, innovation

and employment. Wollongong TCCON and NDACC measurements are supported by the Australian Research Council, grants DP160101598, DP140101552, DP110103118, DP0879468 and LE0668470. The Reunion Island TCCON measurements are supported by Belgian Science Policy through contracts FR/35/IC1 to IC3.





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
