# Peer review of "An inter-comparison of total column-averaged nitrous oxide between ground-based FTIR TCCON and NDACC measurements at seven sites and comparisons with the GEOS-Chem model"

_Atmospheric Measurement Techniques, 2018_

## Referee Comment (RC1) · Anonymous Referee #2 · 26 Dec 2018

The TCCON and NDACC are two well-known international networks based on ground-based solar FTIR instruments. These two networks have observed globally over several years and their observations are extensively used in atmospheric physics and chemistry. The $CH_4$, HCl, HF, CO, $N_2O$ are target species of both networks. A few studies have performed inter-comparison between the TCCON and the NDACC for certain gases, e.g., CO and $CH_4$. However, no inter-comparison between both datasets is available in literature for nitrous oxide ($N_2O$),which is the third most important anthropogenic greenhouse gas in the Earth's atmosphere after carbon dioxide ($CO_2$) and methane ($CH_4$). This study presents a global view of the $XN_2O$ measurement differences between these two networks at seven sites (Ny-Ålesund, Sodankylä, Bremen, Izaña, Reunion Island, Wollongong and Lauder) covering a large latitudinal range from 45.0˚S to 78.9˚N. Furthermore, trends and seasonal cycles of $XN_2O$ derived from the TCCON and NDACC measurements and the nearby surface flask sample measurements were compared with the results from the GEOS-Chem model a priori and a posteriori simulations. I would like to regard the novelty of this paper as moderate since previous studies have performed similar comparisons rather than $N_2O$ but for CO and $CH_4$. However, this work can be a supplement of current understanding and should be in the literature. Generally, this paper is well written, fits well in the scope of AMT, and I recommend for publication with few corrections.

Specific comments.

1. In introduction part, the authors present many descriptions regarding why measuring N2O is important, how it can be measured by both the NDACC and TCCON networks, the usage of these measurements, and how they can be reproduced by CTM models. However, introduction for the key point of the paper, i.e., the scientific goal of the comparison is quite simple. More descriptions should be better, e.g., the authors can briefly introduce the previous comparisons between the TCCON and NDACC measurements for other gases, and what' the findings (optional request).

2. The reason why choose these 7 sites for comparisons because they covered a large latitudinal range from 45.0˚S to 78.9˚N. The global coverage is quite good. It is necessary to include this clarification. Besides, I think section 2 contain much

information and can be more structured, e.g., divide it into two subsections, with one for network description and the other one for theoretical analysis regarding what we can expect from the comparison, or why the difference exists.

3. What do you mean by " only the TCCON measurements from the Bruker 125HR at Lauder are used"? You also used the NDACC measurements, right?

4. Not all acronyms in tables 3 and 4 are defined for the first time, e.g., MWs for the micowindows, TCCONap for TCCON a priori.

5. In table 3 and corresponding text, if the NDACC $N_2O$ total column at Sodankyla is divided into 3 partial columns. The partial DOFS at each partial layer is less than unity, do you notice this?

6. Page7 line 4, there is no definition for VMR.

7. In figure 3, the differences between the TCCON and NDACC measurements at nyalesund and sodankyla are seasonal dependent. However, the differences at all other site are quite low and seasonal independent. The authors presented a detailed analysis for the seasonal dependent difference (sodankyla), but for the seasonal independent difference, the authors did not present any analysis. According to equation (4) and figure 2, the TCCON and NDACC avks are quite different. For the seasonal independent and the lower difference, the TCCONap could be more closer to the true state of the atmosphere, right?

8. In section 4, both nyalesund and sodankyla show seasonal dependent difference. Here the authors only select sodankyla for case study. The reason should be clarified.

9. Page 9 line 3 ,"…low XN2O values in the TCCON measurements in Figure 4 correspond to periods of high PV…". As far as I judge from this figure, it is not always right. Please check if the plotting is correct.

10. Page 9 line 10, ACE-FTS is not the first time and should be defined in previous section.

11. Page 10 line 3, one "rapidly" should be removed.

12. Page 10 line 6. The smoothed ACE-FTS measurements are close to the NDACC retrieved N2O profiles for both inside and outside polar vortex cases, because the NDACC retrieval has a good sensitivity and the NDACC retrieval is able to capture the change in the stratosphere. However, the TCCON retrieval overestimates the deviation from the a priori in the stratosphere. Another reason is that you smoothed the ACE-FTS using the NDACC avk, but if you use the TCCON avk. The smoothed ACE-FTS measurements should also close to TCCON profile.

13. In figure 5, why you use the TCCON a priori profile rather than the scaled TCCON a priori profile (the retrieved ) in comparison. In caption should state the error bars are included.

14. In table 6, I recommend to include the longitude and latitude information of the FTIR site.

15. The agreement in Figure 7 is improved, however, it still shows that the NDACC over/under estimated TCCON at low/high concentration.

---

## Referee Comment (RC2) · Anonymous Referee #2 · 27 Dec 2018

It is hard to distinguish summer or winter season in Figs. 3,4,7. I recommend the authors to inlcude the label month with an interval of 3 or 4 in these plottings.

---

## Referee Comment (RC3) · Anonymous Referee #1 · 14 Jan 2019

The manual is clearly written and well organized. It characterizes in detail and explains NDACC vs. TCCON retrieval performance discrepancies for N2O (dry-air column averaged), along with providing insight into CTM model performance in the SH. Although the conclusions presented in the abstract and conclusions sections are supported by the data shown, there are a number of places where the manuscript lacks specificity. I recommend publication after the issues below are addressed.

[Figure]

Major comments:

Equation 1: SBF stands for? Alpha is a scaling factor, is beta too? Is the functional form of SBF necessary here?

Pg. 5, regularization: "OEM" vs "Tikhonov" is too general, especially since the OEM choice leads to 1 extra DOF in Table 3. Is the OEM prior covariance based on WACCM runs? Is Tikhonov diagonal or with a correlation length?

P7L4: 0.06% is called "relatively small" here for differences in retrievals due to apriori profile, but 0.09% was called "negligible" on P5L24 in regard to differences caused by regularization schemes.

P5L22: you're really testing for 4 things in Table 4: spectroscopy, regularization, window and apriori profile. Stating this early is less confusing.

P7L9: "apart from spectroscopy causing a bias between different retrieval windows" is imprecise and also confusing w.r.t. to the previous statement that spectroscopy is the same (in a given microwindow). I suggest "apart from the different sensitivity of the forward model to the underlying true state in different microwindows, e.g., on account of spectroscopic differences,"

P8L5, regarding Fig3: "bias . . . increases with time" -> I cannot see this effect in Fig3 Fig 4: The lowest TCCON N2O days occur for shades of orange corresponding roughly to April/May, when PV is low (maybe this is the later 2014 anomaly). What do we learn if we color the scatter plot by PV instead of month of year, or by SZA?

Fig 5: Give n=3 and 40 for inside/outside measurements in caption. State the nature of the error bars. This plot is hard to read, consider splitting into two panels.

Section 5: were the NyAlesund and Sodankyla TCCON data a priori-corrected in the model comparisons? If not, how would including this change Fig. 8/9/10?

P14L3: "except at NyAlesund." -> and with very high variability at Sodankyla, which

should be noted here as well.

P14L4: In this paragraph, the TCCON-derived trend is stated as "0.8-0.9 ppb/year" and described as "slightly smaller" than NDACC and flask sample trends of 0.9 – 1.0 ppb/year. I put a horizontal line to Fig 8 and find the highest TCCON trend to be ∼0.85 (Wollongong), with all others being lower. Please calculate a single value of the TCCON, NDACC, flask, GEOS-priori and GEOS-posterior trends and be precise in comparing one to another. On L9: updates to the TCCON profile with a 0.3%/yr growth rate: is that updated only at the surface or through a scaling of the whole profile? Give the precise change in the apriori structure and a precise trend value change.

P14L18-19: "there are no full extent of the minimum . . . at NyAlesund in 2007, 2009, 2011" -> it's hard to see in that figure whether this is due to a lack of data or could it be because polar vortex intrusions were fewer in those years?

P14L20: "comparatively more FTIR data" -> be specific: there is nearly 10X more data at Sodankyla. Also, since this data occurs in the shortest time series, this supports your previous argument about assuming constant growth rates and not worrying about different time series lengths at different TCCON/NDACC stations.

Fig 9: It's hard to argue that there's a maximum in FTIR seasonal variations from Aug-Oct and a minimum from Feb-Apr at NyAlesund since there are no measurements in Feb, Sep, and Oct. This statement only holds true at Sodankyla and only for TCCON data (there's no clear maximum in autumn NDACC data). In this paragraph, the use of "slightly larger" (L6), "much larger" (L7), and "good agreement" (L10) is qualitative and debatable, especially in contrast with the high precision and high accuracy of the NDACC and TCCON data sets that was painstakingly laid out in Section 2. "good agreement" here appears to mean "better than factor of 2", though it is hard to see for Bremen and Izana. Please quantify. Also, I can't see an opposite pattern of seasonal variations at Wollongong as compared to the model; to me it appears rather flat, on average.

P16L4: "below 8 km" -> "from 0 to 8 km" will distinguish it from surface flask measurements better

P16L7: while Wollongong and Lauder may have comparable tropopause heights (check and quantify), they are separated by 10 degrees of latitude and exist in different climates, which should not be dismissed too quickly. For example, Fig 10 third row (8-17 km) clearly shows that if stratospheric processes are responsible for the discrepancy with GEOS-Chem, this is a stronger effect at the southern mid-latitude Lauder rather than the sub-tropical Wollongong.

P18L7: "slightly underestimated" -> please quantify

Minor comments:

P2L12: million -> billion

P3L11: "study is [that] to"

P3L29: "of [the] O2 in [the] dry air"

P5L15: "in these two spectroscopy" -> "in these two spectroscopic databases"

P6L4: "difference. Consequently," -> "difference, consequently,"

P7L18: "are from" -> "range from"

P7L19: "with [the] standard deviations"

P8L10: "autumn" -> autumn

P9L5, L12, L18, L23: "polar vortex" -> "the polar vortex" (also in Fig. 6 and 7 caption and P16L28)

P9L8: "the isolation from mid-latitude refreshing" -> "dynamic confinement"?

P9L13: "[the] mole fractions"

P9L21: "criteria" -> "criterion"

P10L3: "[rapidly] decreases more rapidly"

P10L9: "[and] explaining why"

P14L16: "might be explained by [the lack of measurements;] the fact"

P17, Fig. 10 caption: "second to fourth panels" -> "second to fourth row panels"
* * *

---

## Author Comment (AC1) · 13 Feb 2019

The comment was uploaded in the form of a supplement:
https://www.atmos-meas-tech-discuss.net/amt-2018-340/amt-2018-340-AC1-supplement.pdf

---

## Author Comment (AC3) · 13 Feb 2019

The comment was uploaded in the form of a supplement:
https://www.atmos-meas-tech-discuss.net/amt-2018-340/amt-2018-340-AC3-supplement.pdf

---

## Author Comment (AC2)

**Black: referee's comments red: authors' answers**

*First of all, we want to thank the two referees for the detailed analysis of our paper. For the details, please look into the paper with keeping track of changes.*

**Referee #2**

The TCCON and NDACC are two well-known international networks based on ground-based solar FTIR instruments. These two networks have observed globally over several years and their observations are extensively used in atmospheric physics and chemistry. The CH4, HCl, HF, CO, N2O are target species of both networks. A few studies have performed intercomparison between the TCCON and the NDACC for certain gases, e.g., CO and CH4. However, no inter-comparison between both datasets is available in literature for nitrous oxide (N2O), which is the third most important anthropogenic greenhouse gas in the Earth's atmosphere after carbon dioxide (CO2) and methane (CH4). This study presents a global view of the XN2O measurement differences between these two networks at seven sites (Ny-Ålesund, Sodankylä, Bremen, Izaña, Reunion Island, Wollongong and Lauder) covering a large latitudinal range from 45.0°S to 78.9°N. Furthermore, trends and seasonal cycles of XN2O derived from the TCCON and NDACC measurements and the nearby surface flask sample measurements were compared with the results from the GEOS-Chem model a priori and a posteriori simulations. I would like to regard the novelty of this paper as moderate since previous studies have performed similar comparisons rather than N2O but for CO and CH4. However, this work can be a supplement of current understanding and should be in the literature. Generally, this paper is well written, fits well in the scope of AMT, and I recommend for publication with few corrections.

**Specific comments.**

In introduction part, the authors present many descriptions regarding why measuring N2O is important, how it can be measured by both the NDACC and TCCON networks, the usage of these measurements, and how they can be reproduced by CTM models. However, introduction for the key point of the paper, i.e., the scientific goal of the comparison is quite simple. More descriptions should be better, e.g., the authors can briefly introduce the previous comparisons between the TCCON and NDACC measurements for other gases, and what' the findings (optional request).

Thanks for your suggestions. The scientific goal of the comparison is written as "The target of this study is to better understand the discrepancies between the TCCON and NDACC N2O measurements, and to know whether two networks can be combined with atmospheric chemistry models for evaluation, seasonal cycles and long-term trend analyses." As the introduction is focus on N2O, we prefer to avoid to mention the TCCON and NDACC comparison in CH4 and CO here.

2. The reason why choose these 7 sites for comparisons because they covered a large latitudinal range from 45.0°S to 78.9°N. The global coverage is quite good. It is necessary to include this clarification. Besides, I think section 2 contain much information and can be more structured, e.g., divide it into two subsections, with one for network description and the other one for theoretical analysis regarding what we can expect from the comparison, or why the difference exists.

These sites are selected because "Both TCCON and NDACC N2O measurements are available at these sites in the time period of 2007 - 2017." The introduction of TCCON and NDACC networks has

been carried out in the Section 1. There, we refer to Wunch et al., (2011) and De Maziere et al., (2018) for a detail description of TCCON and NDACC, respectively.

3. What do you mean by "only the TCCON measurements from the Bruker 125HR at Lauder are used"? You also used the NDACC measurements, right?

The TCCON measurements from the Bruker 125HR at Lauder are selected, because we do not want to introduce the uncertainty due to the changing of instrument from the Bruker 120HR to the Bruker 125HR in TCCON data. For the NDACC measurements at Lauder, all the measurements are recorded by the Bruker 120HR during 2007 – 2017. More information can be found in Pollard et al., (2017).

4. Not all acronyms in tables 3 and 4 are defined for the first time, e.g., MWs for the micowindows, TCCONap for TCCON a priori. Added

5. In table 3 and corresponding text, if the NDACC N2O total column at Sodankyla is divided into 3 partial columns. The partial DOFS at each partial layer is less than unity, do you notice this?

As we mentioned in the text "The degrees of freedom for signal (DOFS) at these sites are in the range of 2.4–4.5. The range in DOFS is quite large; while it is known in the NDACC community that the DOFS of N2O retrieval is usually between 2.5-3.5 (Angelbratt et al., 2011; García 20 et al., 2018). The wide range of DOFS in this study does not affect the total column, but we limit to 3 partial columns for NDACC vertical profiles."

For the DOFS less than 3.0, for example 2.4 at Sodankyla, we do not recommend to divided into 3 partial columns. We did not divide into 3 layers for Sodankyla NDACC data in this study.

6. Page7 line 4, there is no definition for VMR. It has already been defined in P2L29 "where 0.2095 is the constant volume mixing ratio (VMR)"

7. In figure 3, the differences between the TCCON and NDACC measurements at nyalesund and sodankyla are seasonal dependent. However, the differences at all other site are quite low and seasonal independent. The authors presented a detailed analysis for the seasonal dependent difference (sodankyla), but for the seasonal independent difference, the authors did not present any analysis. According to equation (4) and figure 2, the TCCON and NDACC avks are quite different. For the seasonal independent and the lower difference, the TCCONap could be more closer to the true state of the atmosphere, right? Yes

8. In section 4, both nyalesund and sodankyla show seasonal dependent difference. Here the authors only select sodankyla for case study. The reason should be clarified. The reason to choose sodankyla is that the co-located ACE-FTS measurements are available above Sodankyla. Since ACE-FTS use the solar occultation mode, there is no co-located ACE-FTS measurement at Ny-Alesund.

9. Page 9 line 3 , "...low XN2O values in the TCCON measurements in Figure 4 correspond to periods of high PV...". As far as I judge from this figure, it is not always right. Please check if the plotting is correct.

Thanks for the suggestion. To make it more clear, the scatter plot is colored with PV value now.

10. Page 9 line 10, ACE-FTS is not the first time and should be defined in previous section. Corrected

11. Page 10 line 3, one "rapidly" should be removed. Corrected

12. Page 10 line 6. The smoothed ACE-FTS measurements are close to the NDACC retrieved N2O profiles for both inside and outside polar vortex cases, because the NDACC retrieval has a good sensitivity and the NDACC retrieval is able to capture the change in the stratosphere. However, the TCCON retrieval overestimates the deviation from the a priori in the stratosphere. Another reason is that you smoothed the ACE-FTS using the NDACC avk, but if you use the TCCON avk. The smoothed ACE-FTS measurements should also close to TCCON profile.

The smoothed ACE-FTS profiles with TCCON avk are still far away from the TCCON retrieved profiles (especially in the stratosphere), because the TCCON AVK is not close to 1.0.

13. In figure 5, why you use the TCCON a priori profile rather than the scaled TCCON a priori profile (the retrieved) in comparison.

We used the TCCON ap profiles, mainly due to 3 reasons: 1) TCCON standard product only provide the XN2O and a priori profile. There is no scaling factor or retrieved profile in the TCCON data. 2) The retrieved N2O profile has the same profile shape as the one from the a priori profile, since GGG2014 perform a profile scaling. 3) The retrieved XN2O is very close to the a priori XN2O (normally within 3 ppm), therefore the retrieved profile is close to the a priori profile.

In caption should state the error bars are included. Added now.

14. In table 6, I recommend to include the longitude and latitude information of the FTIR site. The latitude and longitude information of each FTIR site can be found in Table 1.

15. The agreement in Figure 7 is improved, however, it still shows that the NDACC over/under estimated TCCON at low/high concentration.

Figure 7 shows that the NDACC measurements are slightly larger/lower than TCCON measurements at low/high concentration, but the difference is within the uncertainty from TCCON and NDACC measurements. Therefore, we think the NDACC and TCCON (after a priori correction) XN2O measurements are in a good agreement.

One more comment: It is hard to distinguish summer or winter season in Figs. 4,7. I recommend the authors to inlcude the label month with an interval of 3 or 4 in these plottings.

The plots have been updated. Now the scatters are colored with PV values.

Referecnes:

Pollard, D. F., Sherlock, V., Robinson, J., Deutscher, N. M., Connor, B., and Shiona, H.: The Total Carbon Column Observing Network site description for Lauder, New Zealand, Earth Syst. Sci. Data, 9, 977–992, https://doi.org/10.5194/essd-9-977-2017, 2017.